# Amino Acid Transporters as Targets for Cancer Therapy: Why, Where, When, and How

**DOI:** 10.3390/ijms21176156

**Published:** 2020-08-26

**Authors:** Stefan Bröer

**Affiliations:** Research School of Biology, Australian National University, Canberra ACT 2600, Australia; Stefan.broeer@anu.edu.au; Tel.: +61-2-6125-2540

**Keywords:** LAT1, ASCT2, xCT, SNAT1, SNAT2, mTOR, GCN2, solute carrier

## Abstract

Amino acids are indispensable for the growth of cancer cells. This includes essential amino acids, the carbon skeleton of which cannot be synthesized, and conditionally essential amino acids, for which the metabolic demands exceed the capacity to synthesize them. Moreover, amino acids are important signaling molecules regulating metabolic pathways, protein translation, autophagy, defense against reactive oxygen species, and many other functions. Blocking uptake of amino acids into cancer cells is therefore a viable strategy to reduce growth. A number of studies have used genome-wide silencing or knock-out approaches, which cover all known amino acid transporters in a large variety of cancer cell lines. In this review, these studies are interrogated together with other databases to identify vulnerabilities with regard to amino acid transport. Several themes emerge, such as synthetic lethality, reduced redundancy, and selective vulnerability, which can be exploited to stop cancer cell growth.

## 1. Why

Amino acids are the building blocks of proteins and precursors for a number of essential metabolites, such as C1 compounds, nucleotides, glutathione, polyamines, hexosamines, creatine, and so forth [1,2,3,4] (Figure 1). Amino acids are classified into essential and nonessential amino acids (Table 1). The definition is based on the demand of organisms to grow and thrive, a concept which can also be applied to cancer cells. A metabolically essential amino acid is defined as an amino acid the carbon skeleton of which cannot be synthesized [5]. Removal of such an amino acid from the media halts cancer cell growth. It is instructive to estimate the net amino acid demand of a growing cancer cell and compare it to the transport capacity. In its exponential growth phase, a fast-growing cancer cell line will double its mass every 20 h. The growth is not linear, but equates to about 800 ng protein/min per mg cell protein. Leucine is the most frequently encoded essential amino acid comprising 7.6% of 800 ng (60 ng). Its molecular weight in a protein is 113 Da, which equals 540 pmol of leucine/min per mg protein. The transport capacity for leucine in many cancer cells is much larger than this, namely about 10 nmol/min per mg protein (at a plasma-like concentration of 100 μM), but competition between the amino acids in blood plasma needs to be considered. As discussed below, transport of amino acids is mediated by a mix of uniporters, symporters, and antiporters. The import of 90% of the net amino acid demand is mediated directly or indirectly by SNAT1 (SLC38A1) and SNAT2 (SLC38A2), because they are the main neutral amino acid loaders in cancer cells and provide exchange substrates for antiporters to capture amino acids not transported by SNAT1 and SNAT2 (Figure 2). This would equate to about 6.5 nmol of neutral amino acid influx per min per mg protein through both transporters at plasma concentrations of its substrates (taken together about 1 mM), which is within the experimentally observed range [6]. The highly dynamic regulation of SNAT2 probably tunes its activity/expression to tightly match net demand for cell growth.

Some amino acids are conditionally essential, which indicates that the amino acid is required for specific metabolic activities, which outstrip the capacity of the cell to synthesize it. A well-known example is the glutamine addiction of cancer cells [7], but asparagine [8], arginine [9], and cysteine/cystine [10,11] can also be conditionally essential. While differentiated cells also acquire amino acids, it is important to realize that all cells efficiently recycle amino acids through lysosomal and proteasomal protein digestion [1,12,13]. As a result, the net demand of a differentiated cell for amino acids is very small. Moreover, amino acid restriction in differentiated cells can be compensated by autophagy, however, this recycling process does not provide new building blocks for growing cells. Proteasomal protein recycling is nevertheless an essential function for cancer cells as illustrated by proteasomal inhibitors and compounds that interfere with protein ubiquitination [14], which cause apoptosis due to lack of recycling of cell cycle-regulating proteins [15]. Thus, it stands to reason that there is a significant therapeutic margin between cancer cells and differentiated cells when it comes to curtailing amino acid supply. This is exemplified by asparaginase treatment of childhood lymphoblastic leukemia, a cancer type with negligible asparagine synthetase activity [8]. In the framework of this review, it is important to recognize that asparaginase treatment is used as a combination treatment, not as a monotherapy. In addition to their role as building blocks, amino acids are important signaling molecules that regulate mTORC1 activity, which is crucial for ribosome biogenesis and protein translation [16]. An active mTORC1 is a requirement for cell growth. Amino acids are also important regulators of the integrated stress response [17].

The reliance of cancer cells on specific amino acids is typically tested in vitro using cancer cell lines. Media formulation emulates the composition of blood plasma and is supplemented with fetal calf serum, but it is important to recognize the limitations of analyzing cell growth in media [18]. For instance, fatty acid biosynthesis is an essential pathway in cell culture media, because cancer cells need to synthesize phospholipids to generate more membranes. However, from blood plasma, cancer cells can acquire fatty acids and phospholipids through lipoproteins, lipid exchange factors, and fatty acid transport [19]. Blood plasma also contains metabolites that may affect metabolism. Addition of uric acid to media at plasma concentrations reduces pyrimidine synthesis and sensitivity to 5-fluorouracil [20]. It has also been observed that some cancer cell lines can use phagocytosis of proteins to sustain amino acid supply [21], while other cell lines rather use pinocytosis to activate mTORC1 [22]. Media formulation can also change the cell’s relative reliance on metabolic pathways. High levels of glucose foster glycolysis, while high levels of glutamine drive glutaminolysis. Freshly formulated media often contain cysteine, 98% of which will be converted into cystine within six days. This will shift reliance of glutathione biosynthesis from cysteine to cystine. Unfortunately, there are no universal standards of amino acid composition, which vary substantially between standard media [18]. In vivo, the tumor microenvironment can be modified by depletion or release of nutrients from neighboring cells [23,24]. For instance, glutamine can be provided by cancer-associated fibroblasts and fatty acids can be provided by adipocytes. Alanine is exchanged between pancreatic cancer cells and pancreatic stellate cells [25]. Cancer cells increase tryptophan metabolism to suppress immune cells in the vicinity of a tumor [26].

Some amino acids that are generally considered nonessential, such as glutamine and asparagine, can become essential for cancer cells. Most cancer cells are glutamine-dependent, because of its contribution to glutaminolysis [27], hexosamine biosynthesis, glutathione biosynthesis, proline biosynthesis, and nucleotide biosynthesis (Figure 1). This metabolic adaptation is caused in part by the constant withdrawal of tricarboxylic acid (TCA) cycle intermediates, particularly oxaloacetate to form aspartate [27]. Thus, glutamine acts as an anaplerotic source for the TCA cycle in cancer cells. Some of these roles can be fulfilled by asparagine [28,29], but the use of asparagine to generate glutamine requires a significant flux of acetyl-CoA into the TCA cycle, while glutaminolysis can run in the absence of acetyl-CoA and produces large amounts of aspartate for nucleotide biosynthesis. In agreement with this notion, asparagine was able to prevent cell death after glutamine depletion, but did not restore levels of TCA cycle intermediates [29]. It is important to realize that glutaminolysis is a linear pathway, where removal of intermediates does not compromise its function, unlike the TCA cycle (Figure 1).

In summary, cancer cells have a high demand for amino acids and removal of essential or conditionally essential amino acids from the media halts cell growth. Media are a limited approximation of the tumor environment, which can be modified by use and release of metabolites involving stromal cells in the vicinity of the tumor. Because of the ability to induce autophagy, restriction of essential amino acids stops cell growth, but does not immediately lead to cell death. This limits toxicity for differentiated cells, but also limits the ability to kill already established tumors.

## 2. Where

Amino acid transport in cancer cells is mediated by a set of uniporters, symporters, and antiporters (Figure 2). Only uniporters and symporters can mediate net movement of amino acids into the cell. They serve as amino acid loaders, while antiporters serve to harmonize the amino acid composition in the cytosol [1,6]. The harmonization becomes apparent when an amino acid is depleted in the cytosol and therefore cannot serve as an efflux substrate, but at the same time will be imported in exchange for an abundant cytosolic amino acid. Amino acid transporters accept groups of amino acids such as large neutral (L-AA^0^), small neutral (S-AA^0^), small and medium neutral (S/M-AA^0^), cationic amino acids (AA^+^), or anionic amino acids (AA^−^). As an example, net uptake of glutamine by cancer cells is mediated by SNAT1 and SNAT2 [6].

Cytosolic glutamine can then serve as an exchange substrate for ASCT2, LAT1, and so forth. Glutamine can also be metabolized to glutamate, which serves as an exchange substrate for xCT, to import cystine. Figure 2 only depicts transporters that are found in a large variety of cancer cell lines, thus forming a minimal set of amino acid transporters to sustain cell growth. Notably, cancer cells have very low levels of glutamate transporters, relying instead on glutamine import and glutaminase activity to generate glutamate. Similarly, many cancer cells rely on the neutral amino acid citrulline to generate arginine (Figure 3), which in turn requires aspartate generated by glutaminolysis. Thus neutral amino acid transport is crucial for cancer cells.

For many years, the research community has focused on three transporters, namely ASCT2 (SLC1A5) [30,31], LAT1 (SLC7A5) [32,33], and xCT (SLC7A11) [11]. The main reason for the interest in ASCT2 and LAT1 is the upregulation of these two transporters in many cancer cells [31,34]. The idea was endorsed by the suggestion that ASCT2 and LAT1 in combination were required to activate mTORC1 [35]. The hypothesis stipulated that ASCT2 imports glutamine, which is subsequently used as an exchange substrate to import leucine via LAT1, which in turn activates mTORC1. The flaw of this hypothesis was that ASCT2 is an exchanger [36,37] and thus net accumulation of glutamine through this mechanism is difficult to explain. A substantial body of evidence now demonstrates that this mechanism does not work as suggested. In particular, genomic mutation of ASCT2 in selected cell lines does not prevent cancer cell growth or mTORC1 signaling [6,38,39,40]. This is not too surprising as transporters such as SNAT1 and SNAT2 have the same substrate specificity as ASCT2 and are mediating net uptake of glutamine into cancer cell lines [6]. By contrast, LAT1 appears to be the major if not only pathway for many essential AA (BCAA, aromatic AA) to enter the majority of cancer cells. Surprisingly, pharmacological inhibition and genomic mutation of LAT1 also does not abolish cell growth or mTORC1 activity in some cell lines, suggesting cell-line-specific compensation or large reserve capacity [41,42]. The elevated activity of ASCT2 and LAT1 in cancer cells can be rationalized by their role in ensuring a harmonized amino acid pool in the cytosol to maintain optimal conditions for protein biosynthesis. 

The role of xCT in cancer cell growth has largely been derived from its involvement in ferroptosis. Ferroptosis is cell death caused by extensive lipid peroxidation and can be prevented by iron chelators and lipid antioxidants, but also by addition of 2-mercaptoethanol [43,44]. Mechanistically, these observations are linked through glutathione (GSH). Glutathione is a substrate of glutathione peroxidase 4 (GPX4), a lipid peroxidase, which converts and defuses lipid peroxides to lipid alcohols. Lipid peroxidation can also be prevented by lipid antioxidants, while iron chelators reduce the production of oxygen radicals. In the glutathione tripeptide cysteine is the key residue, which forms a disulfide bridge when oxidized (G-S-S-G). NADPH is required to reduce glutathione disulfide to GSH for another cycle of the reaction. Cells can acquire either cysteine or cystine (Cys-S-S-Cys) for incorporation into GSH. The cytosolic concentration of GSH is 1–2 mM, an amount that needs to be synthesized when cells duplicate. Sulfasalazine is a drug with a variety of effects and targets [45], but its growth-restricting action is often reduced by addition of 2-mercaptoethanol, which converts cystine to cysteine. Accordingly, one of the targets of sulfasalazine is the cystine transporter xCT. Notably, cysteine comprises 20–30% of the total cystine/cysteine pool in plasma [46]. Consequently, cysteine uptake, which can be mediated by ASCT1, ASCT2, SNAT1, and SNAT2 (Figure 1), provides redundancy with regard to cystine uptake via xCT. In media where cysteine spontaneously oxidizes to cystine, or where only cystine is added, xCT function is critical. 

Overlooked targets in cancer metabolism are the mitochondrial amino acid transporters. In a complex genome-wide RNAi screen of RAS-driven tumor formation in *Drosophila*, an ornithine carrier homologue was identified as a gene required for tumor growth [47], possibly pointing to the biosynthesis of arginine or the use of ornithine inside mitochondria (Figure 3). Interestingly, a homologue of the arginine transporter CAT1 was also identified in the same screen, but a functional characterization of the transporter is lacking. A small variant of ASCT2 (ASCT2_var) has recently been proposed as the mitochondrial glutamine transporter [48]. Silencing of ASCT2_var was associated with reduced glutamine uptake into mitochondria and reduced labeling of TCA cycle metabolites. The results are difficult to reconcile with a complete ASCT2ko that would also affect ASCT2_var, but did not affect labeling of TCA cycle intermediates [6].

In the past 10 years, several studies have attempted genome-wide silencing or knock-out to identify essential genes for cancer cell growth. As technology improved, these studies have moved from 12 to several hundred cell lines, allowing a systematic evaluation of individual amino acid transporters as cancer drug targets [49,50,51,52,53]. These very large combined datasets are now available online (https://depmap.org; https://depmap.sanger.ac.uk/). The workflow of each of these studies is similar. Initially, a pool of cells is transfected with shRNA or CRISPR guide RNAs covering the whole genome. The multiplicity of infection is set below 1 to avoid silencing/deletion of more than one gene per cell. Subsequently, the pool is selected for successfully transfected cells. The transfected cells are seeded out and allowed to propagate for some time (2–4 weeks). Cells in which an essential gene has been targeted will fail to grow or even die. Cells in which nonessential genes have been targeted will grow at a normal rate. As a result, cells with reduced expression of essential genes will have disappeared or are significantly underrepresented. Each shRNA or guide RNA is barcoded and sequencing of the pool immediately after successful transfection and at the end of the experiment will reveal shRNAs that have prevailed (and thus enriched compared to the pool) and those that have disappeared. An enrichment/depletion score is generated and used for statistical analysis.

All studies have identified a rather small pool of about 300 pan-cancer essential genes. When these are knocked out, cells experience a loss of fitness or disappear altogether. Protein turnover is a prominent function in this pool as exemplified by ribosome subunits, translation factors, and proteasome subunits. This makes sense, because ribosome assembly is a critical nonredundant process. Similarly, the proteasome has a nonredundant function in protein turnover. Other critical processes are RNA polymerization and splicing. All of these functions have little redundancy and removal of one component prevents cells from propagating. Amino acid transporters do not feature among pan-cancer essential genes, but rather cause cell-specific loss of fitness when removed or silenced (Table 2). This suggests that there must be redundancy among amino acid transporters. The transporters that stand out as potential targets to reduce cancer cell growth are ASCT2 (SLC1A5), LAT1 (SLC7A5), CAT1 (SLC7A1), GC1 (SLC25A22), and SNAT2 (SLC38A2). For these transporters, RNAi-mediated silencing was not sufficient to reduce fitness, while CRISPR-mediated knock-out did to a variable extent. This suggests that these transporters must be homozygously deleted, or fully inhibited to achieve cytostasis. There are notable differences between studies. For instance, in the Broad Institute dataset, CRISPR of ASCT2 resulted in loss of fitness, while in the Sanger Institute dataset, it did not. Notably, xCT (SLC7A11) caused loss of fitness in only 8 out of 769 cell lines. Most likely lack of cystine uptake was complemented by cysteine, which can be acquired via ASCT1, ASCT2, SNAT1, and SNAT2 (Figure 1). The mixed results upon ASCT2 silencing/ko are supported by studies focusing on ASCT2. Reduction of growth was observed in hepatoma [54], lung [55], melanoma [56], prostate [57], colorectal [58], gastric [59], and triple-negative breast cancer cells [60,61]. However, other studies suggest that shRNA may have caused nonspecific toxicity or had off-target effects [39], inhibitors used to block ASCT2 were nonspecific [6,62] or did not target ASCT2 at all [42]. The stronger reliance of cancer cells on LAT1 vs. ASCT2 can be rationalized by the ability of LAT1 to transport branched-chain amino acids and aromatic amino acids, which are indispensable. While ASCT2 transports methionine and threonine, these can also be accumulated by SNAT1, SNAT2, and ASCT1 (threonine only). The conditionally essential amino acids glutamine, asparagine, and cysteine are substrates of the same group of transporters. As a result, there is broad redundancy around ASCT2, explaining its nonessential role for in vitro growth of cancer cells [6]. However, genomic deletion of ASCT2 does reduce growth of xenografts by mechanisms that could be related to the tumor microenvironment [40]. This highlights some limitations arising from the use of cell culture media. For instance, ASCT2 may act as an indirect amino acid sensor, causing cells to move to sites with better nutrient availability [38].

## 3. When

There are typically three windows of treatment in cancer therapy. The first is post-surgically, the second is after relapse of a cancer often involving a population of drug-resistant cells, and the third is late-stage cancer where some of the cancer can be removed surgically but other solid cancers require chemotherapy. Solid tumors release cancer cells into circulation with numbers varying widely between five and several hundred cells/10 mL blood [63]. Circulating tumor cells explain why surgery is often followed up by chemotherapy. The presence of circulating tumor cells is correlated with poor outcomes and metastatic spreading [64]. Circulating tumor cells need to extravasate and invade into other tissues, a process which requires expression of a different set of genes than normal growth of a tumor. Currently very few cancer treatment strategies have explored the possibility of post-surgical treatment with the aim of limiting extravasation and tissue invasion [65]. After extravasation, tumor cells remain in the vicinity of blood vessels to optimize nutrient access [66]. Amino acid depletion and inhibition of amino acid transporters can be readily studied in two-dimensional culture or spheroids. More complex tests are available for matrix-dependent invasion and motility. Xenografts in mice also do not account for these more complex features of tumor cells, because the growth of the tumor is mainly monitored at the injection site. It is also important to note that growth inhibition is typically measured using a thinly seeded starting culture and not in an already established cell layer. Similarly, xenografts are followed from the time of injection. All of these factors contribute to the fact that neither growth inhibition in cell culture nor reduced growth of xenografts is a good indicator of success in cancer treatment [67]. Unfortunately, few data are available to assess the role of amino acid transporters in these more complex processes. One example is the prevention of metastasis by sulfasalazine, a nonspecific drug which inhibits xCT. In this experiment, formation of metastases in liver after tail injection of KYSE150 esophagus cancer cells was almost completely prevented by concomitant treatment with sulfasalazine [68]. A recent screen with a sample of 810 mutant mouse strains to identify regulators of metastatic colonization did not include homozygous knock-outs of amino acid transporters [69]. A genome-wide RNAi screen of regulators of endothelial cell migration showed impaired migration upon silencing of EAAT1 (SLC1A3) and PAT1 (SLC36A1) and accelerated migration upon silencing of the amino acid transporter trafficking subunit 4F2hc (SLC3A2) [70].

## 4. How

Several concepts have emerged that can be applied to amino acid transporters as well as other targets, such as (i) reducing redundancy, (ii) selective vulnerability, and (iii) synthetic lethality. Reducing redundancy is a strategy where inhibition of one target is combined with blocking mechanisms that upregulate compensatory pathways. Selective vulnerability refers to specific expression patterns that render a cell dependent on a single transporter. Synthetic lethality refers to the parallel inhibition of two targets which show significant synergy with regards to curtailing cancer cell growth.

Inhibition of LAT1 (SLC7A5) serves as an example of redundancy and selective vulnerability. The function of LAT1 can be replaced by LAT2 (SLC7A8) or a combination of LAT3 (SLC43A1) (BCAA) plus TAT1 (SLC16A10) (aromatic AA). In colon cancer cell lines, those with low expression of LAT2 are typically sensitive to LAT1 knock-out (e.g., RKO, HCC-78), while cell lines with higher levels of LAT2 are typically resistant (HT55, SW620). This information can be derived by analyzing expression data from Garnett et al. [71] or Barretina et al. [72] and analyzing them in combination with knock-out data from the Sanger Cancer Dependency Map [51]. One reason why ASCT2 is dispensable in a number of cell lines is the overlapping substrate specificity with SNAT1 and SNAT2, both of which are highly expressed in essentially every cancer cell line [1]. Many cell lines show significant plasticity and can functionally replace inhibited transporters by upregulation of related transporters through activation of GCN2 [38] (Figure 1). However, selective vulnerability with regard to ASCT2ko may occur in KRAS-driven cancer cells [73]. Redundancy can also occur between amino acid transporters and biosynthetic pathways. For instance, cells expressing high levels of argininosuccinate synthetase (ASS) can synthetize arginine from citrulline, a substrate of LAT1 [74], and may not require a cationic amino acid transporter (Figure 3). Vice versa, cells that show a genetic signature of arginine auxotrophy are more likely to be vulnerable to inhibition of CAT1 (SLC7A1). For instance, LS-411N colon carcinoma cells, SU-DHL-5 B-cell lymphoma cells, and SNU-1 gastric carcinoma cells are sensitive to CAT1 deletion most likely because of low levels of ASS expression combined with low expression levels of y^+^LAT2 (SLC7A6). Arginine auxotrophy is frequently observed in melanoma, hepatocellular carcinoma, prostate cancer, and pancreatic cancer and can be exploited using arginine depletion [75]. Redundancy can be limited by concurrent targeting of GCN2, which showed significant synergy together with ASCT2ko [38] and asparaginase treatment [17,76]. However, GCN2 may not be the only pathway involved in transporter plasticity [77]. 

Another case of selective vulnerability are estrogen-receptor (ER)-positive breast cancer cells, which upregulate the general amino acid transporter ATB^0,+^ (SLC6A14) [78]. Prolonged treatment of ER-positive cell lines with α-methyl-DL-tryptophan induced apoptosis, but it is unclear whether the compound has additional targets. Notably, ER-positive cell lines did not show loss of fitness upon ATB^0,+^ ablation (Table 2).

The cystine/glutamate exchanger xCT provides cancer cells with cysteine for glutathione biosynthesis. It has been proposed that a combination of treatments that deplete glutathione can be used to target cancer cells, in particular cells with p53 mutations that appear to reduce xCT expression [79]. Depletion of glutathione makes cells vulnerable to ferroptosis [43]. Depletion of glutathione by APR-246, which binds thiol-groups, in combination with inhibition of xCT could make cancer cells more vulnerable to oxidative stress. Accordingly, it may be necessary to increase oxidative stress to facilitate cell death in xCT ko cells [80,81]. This is consistent with the absence of xCT in genome-wide ko screens. Inhibition of xCT may contribute to the anticancer activity of tyrosine-kinase inhibitor sorafenib [43]. Promisingly, the combination treatment consisting of the xCT inhibitor erastin and the autophagy inducer temozolomid showed significant synergistic toxicity in glioma cells [82]. Erastin also has multiple targets, in particular the mitochondrial voltage-dependent anion channel, which contributes to its anticancer activity [83]. The expression of xCT varies widely between cancer cells and expression levels are not consistent within a given tissue [1]. Certain cancers, such as pancreatic ductal adenocarcinoma, may show selective vulnerability with regard to xCT [84]. There is also a link between glutaminolysis and ferroptosis, providing glutamate as an exchange substrate for xCT and also for glutathione biosynthesis [85,86] (Figure 1). Ferroptosis highlights the critical role of NADPH in cancer cells. Synergy could potentially be generated by blocking NADPH-generating pathways such as the pentose–phosphate pathway, malic enzyme [87,88], or isocitrate dehydrogenase [89]. Synergy has also been found between xCT and anti-PDL-1 check point blockade [90], pointing to a combined use with immunotherapy.

Combination treatment could be useful in cases where transporters contribute to amino acid import and signaling. SNAT2 [91], PAT1, and PAT4 [92,93] are the most promising candidates. It remains to be shown whether inhibition of these transporters would be synergistic with blocking other growth-promoting pathways, such as Ras-Raf-MEK-ERK. Inhibition of mTORC1 could also be meaningful in combination with amino acid transporters that do not affect mTORC1 signaling.

The pharmacology of amino acid transporters has made progress, but high-affinity and selective inhibitors remain scarce (Table 3). As outlined above, use of nonspecific inhibitors has caused overinterpretation of results with regard to reliance of cancer cells on specific transporters. The ferroptosis-related compounds erastin and sorafenib [94] are good examples for anticancer activity generated by acting on multiple targets. The experimental anticancer compound V-9302 also derives its potency from blocking multiple targets [42]. 

## 5. Conclusions

Amino acid transporters, notably LAT1, where clinical trials are ongoing [133], can be targets for cancer therapy, but should be directed at cancers which show selective vulnerability. Unfortunately, tissue of origin cannot substitute tumor expression data in the elucidation/detection of selective vulnerabilities. It is more likely that amino acid transporters are considered in combination with other targets. Targeting synthetic vulnerabilities is just emerging as an intensive area of research. Screens using cell lines with transporter knock-outs can be used to identify such targets. Other synergistic targets may be found in the area of plasticity and adaptation, growth factor signaling, or oxidative stress defense mechanisms. Amino acid transport inhibitors are promising when used in conjunction with other chemotherapeutic compounds, similar to the use of asparaginase.

## Figures and Tables

**Figure 1 ijms-21-06156-f001:**
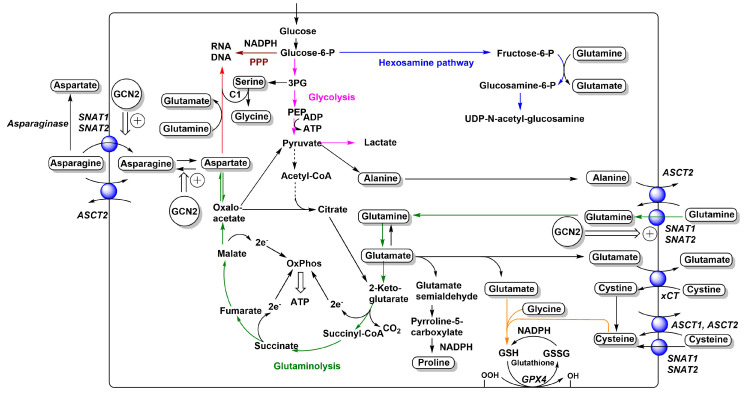
Relationships between neutral and anionic amino acid transporters and metabolism. The main amino acid-related metabolic fluxes in cancer cells, such as glycolysis (pink), glutaminolysis (green), nucleobase biosynthesis (red), hexosamine pathway (blue), glutathione biosynthesis (orange), and pentose–phosphate pathway (brown), are shown. Amino acids are depicted in rounded frames. Transporters that feed into these pathways are depicted as blue spheres and their names are shown in italics. Adaptive changes caused by GCN2 activation are shown as double arrows. Reduced flux of carbon from pyruvate to acetyl-CoA is indicated by a dashed arrow. For more detailed depictions of transport mechanisms, see Figure 2.

**Figure 2 ijms-21-06156-f002:**
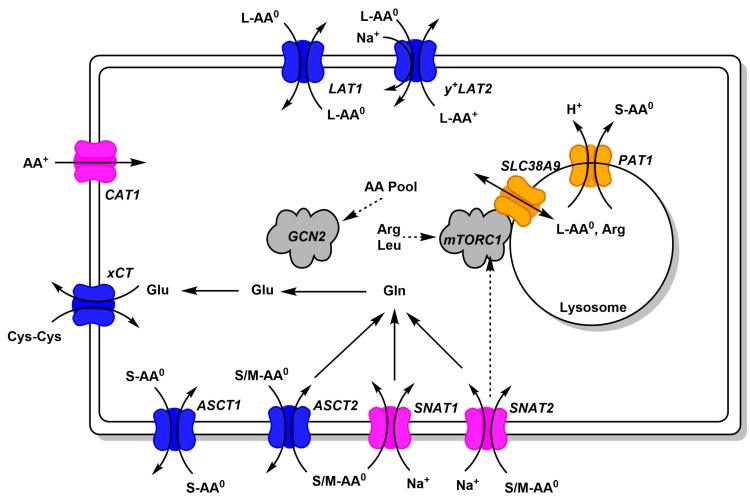
Common amino acid transporters in cancer cells. Amino acid transporters, which are abundant in cancer cells, are shown. Antiporters (harmonizers) are shown in blue, uniporters and symporters in magenta (loaders), and lysosomal transporters in orange. Substrate specificity is indicated using AA^0^ (neutral amino acids) and AA^+^ (cationic amino acids, arginine, lysine, ornithine). Size of neutral amino acids is indicated as small (S, glycine, alanine, serine, cysteine, proline), medium (M, threonine, asparagine, glutamine), large (L, leucine, isoleucine, valine, methionine, phenylalanine, histidine, tryptophan, tyrosine). Links to signaling pathways are shown by dashed arrows.

**Figure 3 ijms-21-06156-f003:**
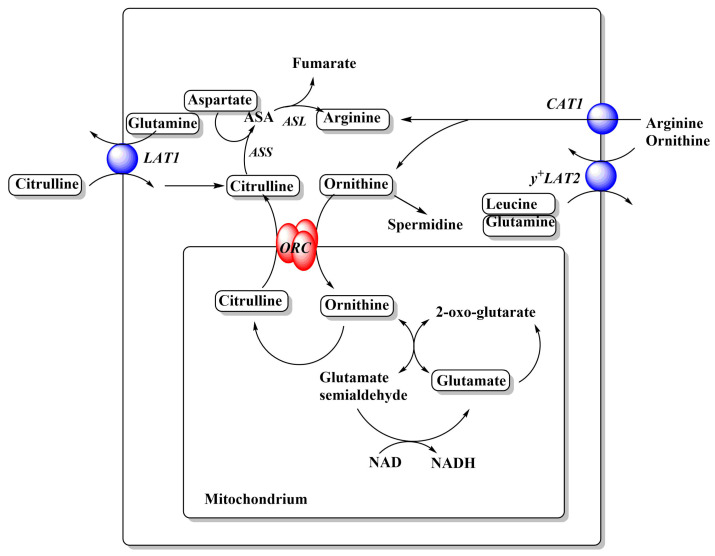
Relationships between cationic amino acid transporters and metabolism. Arginine and ornithine metabolism in cancer cells. A full urea cycle is not expressed in cancer cells. Amino acids are depicted in rounded frames. Transporters that feed into these pathways are depicted as blue spheres and their names are shown in italics. For more detailed depictions of transport mechanism, see Figure 2. Abbreviations: ASS, arginine succinate synthetase; ASA, argininosuccinate; ASL, argininosuccinate lyase; ORC, ornithine carrier.

**Table 1 ijms-21-06156-t001:** Essential and nonessential amino acids in cancer cells.

Amino Acid	Dependence	Comment
Glycine	Nonessential	Mostly generated from serine
Alanine	Nonessential	Mostly generated from pyruvate
Proline	Nonessential	Mostly generated from glutamine/glutamate
Leucine	Essential	Can be generated from ketoisocaproate
Isoleucine	Essential	Can be generated from 2-oxo-3-methylvaleric acid
Valine	Essential	Can be generated from ketoisovalerate
Phenylalanine	Essential	Precursor for tyrosine
Tryptophan	Essential	
Histidine	Essential	
Tyrosine	Nonessential	Requires phenylalanine as a precursor
Aspartate	Nonessential	Mostly generated from asparagine or oxaloacetate
Glutamate	Nonessential	Mostly generated from glutamine or α-ketoglutarate
Arginine	Conditionally essential	Arginino-succinate synthetase-dependent
Lysine	Essential	
Serine	Nonessential	Mostly generated from 3-phosphoglycerate
Threonine	Essential	
Cysteine	Nonessential	Requires methionine and serine as a precursor
Methionine	Essential	Can be generated from homocysteine
Asparagine	Conditionally essential	Asparagine synthetase-dependent
Glutamine	Conditionally essential	Can be generated from glutamate

**Table 2 ijms-21-06156-t002:** Transporters as cancer therapeutic targets. Silencing and CrispR ko data from https://Depmap.org are shown. Transporters are listed by solute carrier number and common name. Four widely used nontransporter targets are listed for comparison at the end of the table. Transporters that are expressed consistently across all cell lines are shown as (+) for low expression levels, (++) for elevated expression levels, (+++) for high expression levels. N.d. not determined.

Solute Carrier	Common Name	Cell Lines Affected by CrispR	Cell Lines Affected by Silencing	Pan-Cancer Expression
SLC1A1	EAAT3	1/769	0/710	
SLC1A2	EAAT2	3/769	0/712	
SLC1A3	EAAT1	1/769	4/671	
SLC1A4	ASCT1	1/769	n.d.	+
SLC1A5	ASCT2	176/769	5/710	+++
SLC1A6	EAAT4	4/769	0/547	
SLC1A7	EAAT5	0/769	1/547	
SLC3A1	rBAT	0/769	6/547	
SLC3A2	4F2hc	335/769	2/547	++
SLC6A5	GlyT2	0/769	0/547	
SLC6A7	PROT	6/769	0/547	
SLC6A9	GlyT1	2/769	0/547	
SLC6A14	ATB^0,+^	0/721	14/711	
SLC6A15	B^0^AT2	1/769	0/708	
SLC6A17	NTT4/B^0^AT3	7/769	0/547	
SLC6A18	XT2/B^0^AT3	0/769	0/710	
SLC6A19	B^0^AT1	3/769	27/712	
SLC6A20	SIT1	0/769	0/708	
SLC7A1	CAT-1	135/769	18/711	+
SLC7A2	CAT-2	0/769	9/501	
SLC7A3	CAT-3	0/721	0/547	
SLC7A4	CAT-4	2/769	0/501	
SLC7A5	LAT1/4F2hc	416/769	1/547	+++
SLC7A6	y^+^LAT2/4F2hc	5/769	0/285	++
SLC7A7	y^+^LAT1/4F2hc	0/769	1/708	
SLC7A8	LAT2/4F2hc	1/769	1/712	
SLC7A9	b^0,+^AT/rBAT	1/769	0/711	
SLC7A10	Asc-1/4F2hc	2/769	0/547	
SLC7A11	xCT/4F2hc	8/769	2/710	++
SLC7A13	AGT1/rBAT	4/769	0/547	
SLC7A14	c	0/769	12/605	
SLC16A10	TAT1	1/769	0/547	
SLC17A6	VGLUT2	1/769	0/547	
SLC17A7	VGLUT1	2/769	5/547	
SLC17A8	VGLUT3	0/769	1/547	
SLC25A2	ORC2	0/769	4/547	
SLC25A12	AGC1	1/769	0/710	
SLC25A13	AGC2	8/769	0/710	
SLC25A15	ORC1	0/769	4/712	
SLC25A18	GC2	24/769	0/501	
SLC25A22	GC1	99/769	0/710	
SLC25A44	BCAA	0/769	0/343	
SLC32A1	VIAAT	1/769	1/547	
SLC36A1	PAT1	0/769	11/710	
SLC36A2	PAT2	0/769	2/547	
SLC36A3	PAT3	0/769	0/547	
SLC36A4	PAT4	0/769	0/547	
SLC38A1	SNAT1	13/769	0/710	+++
SLC38A2	SNAT2	161/769	1/547	+++
SLC38A3	SNAT3	n.d.	1/547	
SLC38A4	SNAT4	1/769	0/547	
SLC38A5	SNAT5	3/721	0/547	
SLC38A6	SNAT6	2/769	1/710	+
SLC38A7	SNAT7	2/769	17/547	+
SLC38A8	SNAT8	0/769	n.d.	
SLC38A9	SNAT9	0/769	16/343	+
SLC38A10	SNAT10	9/769	1/597	
SLC38A11	SNAT11	13/769	1/597	
SLC43A1	LAT3	0/769	2/547	+
SLC43A2	LAT4	0/769	n.d.	
SLC43A3	EEG	1/796	0/547	
**Nontransporters**				
DHFR		520/769	0/710	
NRAS		49/769	31/712	
BRAF		76/769	45/712	
MTOR		767/769	347/712	

**Table 3 ijms-21-06156-t003:** Selectivity and affinity of amino acid transporter inhibitors. Selectivity is listed against other amino acid transporters where tested. A (+) sign indicates that the compound has significant nontransporter targets where known. N.d. not determined; n.a. not available.

Target	Inhibitor	Selectivity	Affinity (pIC_50_)	Reference
SLC1A1	NBI-59159	Selective for EAAT3	7.1	[95]
	DL-TBOA	All EAATs	5.1	[96]
SLC1A2	WAY-213613	Selective for EAAT2	7.1	[97]
	DL-TBOA	All EAATs	5.1	[96]
SLC1A3	UCPH-101	Selective for EAAT1	6.9	[98]
	DL-TBOA	All EAATs	4.1	[96]
SLC1A4	n.a.			
SLC1A5	Benzylserine	Also LAT1, SNAT1, SNAT2	3.0	[6,99]
	γ-glutamyl-p-nitroanilide (GPNA)	Also LAT1, SNAT1, SNAT2	4.0	[6,100]
	V-9302	LAT1, SNAT2, not ASCT2	5.0	[42,61]
	Serine-biphenylmethyl carboxylate	n.d.	4.5	[101]
	(R)-γ-(4-biphenylmethyl)-l-proline	n.d.	5.5	[102]
	Proline and serine-based sulfonic acids/sulfonamides	n.d.	5	[103]
	1,2,3 dithiazoles	n.d.	5.5	[104]
	Compound 10	n.d.	4.0	[105]
SLC6A5	Org 25543	Selective	7.8	[106]
SLC6A7	Compound 58	Selective	7.7	[107]
	LP-403812	Selective	7.0	[108]
SLC6A9	Bitopertin	Selective	7.5	[109]
	R-NFPS	Selective	8.5-9.1	[110]
	SSR 103800	Selective	8.7	[111]
	LY2365109	Selective	7.8	[112]
	GSK931145	Selective	7.6	[113]
SLC6A14	α-methyl-DL-tryptophan	n.d.	3.6	[114]
SLC6A17	n.a.			
SLC6A19	Cinromide	Selective (+)	6.4	[115]
	Nimesulide	Selective (+)	4.6	[116]
	Benztropin	Selective (+)	4.4	[117]
	E4, E18, CB3	Selective		[118]
SLC6A20	n.a.			
SLC7A1	n.a.			
SLC7A2	n.a.			
SLC7A3	n.a.			
SLC7A4	n.a.			
SLC7A5	JPH203	Selective		[119]
	BCH	All System L transporters		[120]
	KMH-233	Selective		[121]
	SKN103	Selective		[122]
	(Z)-4-chloro-*N*-(4-(trifluoromethoxy)phenyl)-5H-1,2,3-dithiazol-5-imine	n.d.		[123]
	2-amino-4-(3,5-Dichloro-phenyl)-butyric acid	n.d.		[124]
SLC7A6	n.a.			
SLC7A7	n.a.			
SLC7A8	BCH	All System L transporters		[120]
SLC7A9	n.a.			
SLC7A10	BMS-466442	Selective	7.4	[125]
	(+)-amino(1-(3,5-dichlorophenyl)-3,5-dimethyl-1H-pyrazol-4-yl)acetic acid (ACPP)		4.0	[126]
SLC7A11	Sulfasalazin	Selective (+)	3.8	[45]
	Sorafenib	Selective (+)	5	[127]
	Erastin	Selective (+)	6.7	[127]
SLC7A12	n.a.			
SLC7A13	n.a.			
SLC7A14	n.a.			
SLC16A10	n.d.			
SLC17A6	n.a.			
SLC17A7	n.a.			
SLC17A8	n.a.			
SLC25A2	n.a.			
SLC25A12	n.a.			
SLC25A13	n.a.			
SLC25A15	n.a.			
SLC25A18	n.a.			
SLC25A22	n.a.			
SLC32A1	vigabatrin		2.1	[128]
SLC36A1	Sertraline		2.4	[129]
SLC36A2	None			
SLC36A3	None			
SLC36A4	None			
SLC38A1	MeAIB	SNAT1,2,4, PAT1	4	[130]
SLC38A2	MeAIB	SNAT1,2,4, PAT1	4	[130]
SLC38A3	n.a.			
SLC38A4	MeAIB	SNAT1,2,4 PAT1		[131]
SLC38A5	n.a.			
SLC38A6	n.a.			
SLC38A7	n.a.			
SLC38A8	n.a.			
SLC38A9	n.a.			
SLC38A10	n.a.			
SLC38A11	n.a.			
SLC43A1	ESK246	Selective	5.0	[132]
	BCH	All System L transporters		[120]
SLC43A2	BCH	All System L transporters		[120]

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
