# Peer review of "Amino Acid Transporters as Targets for Cancer Therapy: Why, Where, When, and How"

_ijms, 2020, doi:10.3390/ijms21176156_

Round 1
Reviewer 1 Report
This review summarizes recent findings about the amino acid transporters, especially in the field of oncology. The manuscript contains cutting-edge knowledges with clear illustrations, and is well-written on the whole. I have just one question regarding LAT1.
P5 line146~ and Fig.2
Do you mean that glutamine is not required for LAT1 to incorporate large neutral amino acids? If so, which amino acid (L-AA0 in fig.2) is excreted by LAT1 for its influx activity? Although I would leave the decision on whether to include this in the manuscript up to the author, I would like to know the author’s view.
Author Response
Many thanks for the kind comments:
Glutamine can be an exchange substrate for the influx of large neutral amino acids via LAT1, but any large neutral amino acid that is a substrate of LAT1 can exchange against any other large neutral amino acid (e.g. histidine vs isoleucine, methionine vs phenylalanine etc). The flaw of the Nicklin hypothesis is in the assumption that ASCT2 mediates the glutamine import. To clarify, I have rephrased:
The hypothesis stipulated that ASCT2 imports glutamine, which is subsequently used as an exchange substrate to import leucine via LAT1, which in turn activates mTORC1. The flaw of this hypothesis was that ASCT2 is an exchanger [34,35] and thus net accumulation of glutamine through this mechanism is difficult to explain.
Reviewer 2 Report
Comments
The review is well written and is very interesting because it furnishes an overview of amino acid transproters in cancer.
Only few comments arose which are listed below:
In Table 1, the precursor of glutamine is missing.
In fig. 1, remove ASCT2 as a cysteine transporter
Lines 93-96; I would suggest to add a comment on the fact that asparagine/glutamine substitution is not straightforward as highlighted in the paper by Krall Nat communic 2016 and Zhang Molecular Cell 2014
Given the importance of NADPH, I would suggest to add a comment on pathways responsible for its synthesis.
In fig. 3 the transport of citrulline by LAT1 lacks a reference. Futhermore, given that glutamine utilization implies the mitochondria, I would suggest to add a comment on the existance of a glutamine transporter in mitochondria, recently identified as a saller isoform of ASCT2 (Yoo et al 2019 cell metabolism)
Author Response
The review is well written and is very interesting because it furnishes an overview of amino acid transproters in cancer.
Response: Thanks for the kind comment.
Only few comments arose which are listed below:
1) In Table 1, the precursor of glutamine is missing.
Response: Thanks for spotting, I have added: Can be generated from glutamate
2) In fig. 1, remove ASCT2 as a cysteine transporter
Response: I would respectfully disagree. We have unpublished MS data showing bidirectional transport of cysteine via ASCT2. Cysteine transport via ASCT2 has also been demonstrated by: Scopelliti AJ, Font J, Vandenberg RJ, Boudker O, Ryan RM. Structural characterisation reveals insights into substrate recognition by the glutamine transporter ASCT2/SLC1A5. Nat Commun. 2018;9(1):38. Published 2018 Jan 2. doi:10.1038/s41467-017-02444-w
3) Lines 93-96; I would suggest to add a comment on the fact that asparagine/glutamine substitution is not straightforward as highlighted in the paper by Krall Nat communic 2016 and Zhang Molecular Cell 2014
Response: Comment much appreciated, I have added: Some of these roles can be fulfilled by asparagine [28,29], but the use of asparagine to generate glutamine requires a significant flux of acetyl-CoA into the TCA cycle, while glutaminolysis can run in the absence of acetyl-CoA and produces large amounts of aspartate for nucleotide biosynthesis. In agreement with this notion, asparagine was able to prevent cell death after glutamine depletion, but did not restore levels of TCA cycle intermediates [29].
4) Given the importance of NADPH, I would suggest to add a comment on pathways responsible for its synthesis.
Response: This is a pertinent suggestion, I have added: Ferroptosis highlights the critical role of NADPH in cancer cells. Synergy could potentially be generated by blocking NADPH generating pathways such as the pentose-phosphate pathway, malic enzyme [85,86] or isocitrate dehydrogenase [87].
5) In fig. 3 the transport of citrulline by LAT1 lacks a reference.
Response: Inserted as suggested: For instance, cells expressing high levels of argininosuccinate synthetase (ASS) can synthetise arginine from citrulline, a substrate of LAT1 [73], and may not require a cationic amino acid transporter (Fig. 3)
6) Futhermore, given that glutamine utilization implies the mitochondria, I would suggest to add a comment on the existance of a glutamine transporter in mitochondria, recently identified as a smaller isoform of ASCT2 (Yoo et al 2019 cell metabolism)
Response: Yes, this is an interesting suggestion, but not easily reconciled with other data. I have added: A small variant of ASCT2 has recently been proposed as the mitochondrial glutamine transporter [48]. Silencing of ASCT2_var was associated with reduced glutamine uptake into mitochondria and reduced labelling of TCA cycle metabolites. The results are difficult to reconcile with a complete ASCT2ko that would also affect ASCT2_var, and did not affect labelling of TCA cycle intermediates [6].